# Photocatalytic Treatment of Methyl Orange Dye Wastewater by Porous Floating Ceramsite Loaded with Cuprous Oxide

**Yue Cheng *** , **Ting Cao, Zhiguo Xiao, Haijie Zhu and Miao Yu**

School of Material Science and Engineering, Jingdezhen Ceramic Institute, Jingdezhen 333403, China;
caot15179802872@163.com (T.C.); xzg@email.ncu.edu.cn (Z.X.); jie15879461745@163.com (H.Z.);
yumiao5345@boe.com.cn (M.Y.)
* Correspondence: cy_jci@163.com

**Abstract:** It is well known that water treatment of printing and dyeing wastewaters is problematic. In order to decompose dyes from dyestuff wastewater and convert them into almost harmless substances for the natural environment, an easily prepared, efficient, practical, and easy-to-regenerate composite material was produced from porous floating ceramsite loaded with cuprous oxide (PFCC). The PFCC samples were prepared and characterized by X-ray diffraction spectroscopy (XRD), scanning electron microscopy (SEM), and energy dispersive spectroscopy (EDS). The material was applied for photocatalytic degradation of methyl orange (MO) in water. The results show that the maximal degradation rate of MO was 92.05% when the experimental conditions were as follows: cuprous oxide loading rate of 8%, PFCC dosage of 20 g/L, the reaction time of 2 h, pH value of 8, and solution initial concentration of 30 mg/L. The degradation processes of MO fits well with the Langmuir–Hinshelwood model in reaction kinetics, and the Freundlich model in reaction thermodynamics, respectively. The degradation mechanism of MO was considered from two perspectives—one was the synergetic effect of adsorption and photocatalytic oxidation, and the other was the strong oxidation of hydroxyl radicals produced by photocatalysts.

**Keywords:** dyestuff wastewater; photocatalysis; methyl orange; cuprous oxide; adsorption; ceramsite

## 1. Introduction

China is the world's largest country in terms of the production and consumption of dyes, as well as the amount of dye and printing wastewater, and pollution from dyeing wastewater is continuing to increase [1,2]. Dye and printing wastewater and dyeing wastewater are among the most problematic industrial wastewaters. Dye wastewaters are emitted in large volumes, and they are distinguished by high chroma and complex chemical composition, and they are not suitable for effective biochemical degradation. The main methods for the detection of pollutants in this type of wastewater are BOD (biochemical oxygen demand) analysis, COD (chemical oxygen demand) analysis, determination of organic toxic substances, and chromaticity [3,4]. Dye and printing wastewater often use filtration, precipitation, coagulation, and physicochemical methods [5]. Photocatalytic oxidation degradation of dye wastewater is a recent research hotspot [6,7]. Studies have shown that using $TiO_2$, $ZnO$, $CdS$, etc., as photocatalysts can effectively degrade organic substances such as dyes in wastewater [8]. However, there are relatively few studies on using cuprous oxide as a photocatalyst [9].

$Cu_2O$ is a *p*-type semiconductor with a band gap of 2–2.2 eV [10]. It can absorb most visible light (wavelengths less than 600 nm) [11]. The theoretical photoelectric conversion efficiency is 18% [12]. Moreover, $Cu_2O$ has abundant sources, is non-toxic, and possesses special photoelectric properties, making it a potential photocatalytic material. The practical value of powders in photocatalytic applications is reduced due to their recovery problems.

Ceramics have good mechanical, chemical, and thermal resistance. Recently, several studies have focused on the development of ceramic carriers with light weight and high

activity. Porous ceramic materials possess many pores on their surfaces and inside their structures, resulting in high porosity, excellent absorption capacity, and high specific surface area. In particular, ceramic foams are regarded as a good candidate for simultaneously providing both aerobic and anoxic zones, due to their unique pore structure [13]. Liu et al. [14] prepared lightweight ceramsite (density 0.80–0.90 g/cm$^3$) with good performance from iron ore tailings. In general, the surface of inorganic materials is negatively charged in water due to the presence of many hydroxyl bonds. However, the surface of bacterial cells and organic pollutants is usually negatively charged in aqueous solution, leading to a repulsive electrostatic interaction between cells and pollutants and carrier surfaces [15]. Surface modification of inorganic materials with iron oxides has been extensively investigated and proven to be effective for the removal of heavy metals [16,17] and organic contaminants from water [18–20]. Additionally, Johnson and Logan [21] found that coating quartz with iron oxide increased bacterial retention by 160%, relative to uncoated quartz.

The purpose of this study was to treat the anionic dye methyl orange (MO) by porous floating ceramsite (PFC) loaded with a cuprous oxide photocatalyst. The prepared loaded photocatalyst has many advantages. Since the density of the ceramsite after the photocatalyst is loaded is close to that of water, it is ensured that the ceramsite will be suspended in the water and in full contact with the dye wastewater, and the sunlight can also be better utilized to realize energy-saving and efficient visible light catalytic degradation of low-concentration organic matter.

## 2. Materials and Methods

### 2.1. Required Materials and Reagents

Black porous floating ceramsite was purchased from Jiangxi Zhiqiang Ceramics Co., Ltd., Jingdezhen, Jiangxi, China. Its bulk density is 293 kg/m$^3$, the cylinder compressive strength is 1.15 MPa, the 1 h water absorption rate is 15.2%, the porosity is 60–65%, the softening coefficient is 0.83, and the average particle size coefficient is 1.5. Copper (II) nitrate hydrate (Cu(NO$_3$)$_2$·3H$_2$O) was purchased from Tianjin Fengchuan Chemical Reagent Co., Ltd., Tianjin, China. Ascorbic acid (C$_6$H$_8$O$_6$) and Sodium hydroxide (NaOH) were purchased from China Pharmaceutical Group Chemical Reagents Co., Ltd., Shanghai, China. Methyl orange (C$_{14}$H$_{14}$N$_3$NaO$_3$S) was purchased from Pharmaceutical Group Chemical Reagents Co., Ltd., Shanghai, China. Hydrofluoric acid (HF) was purchased from Nanchang Xinguang Fine Chemical Plant. Anhydrous ethanol (C$_2$H$_6$O) was purchased from Shanghai Reagent Factory No. 1. Hydrochloric acid (HCl) was purchased from Nanchang Ganjiang Muriatic Acid Chemical Plant. All chemical reagents were analytical reagents and were not purified further.

### 2.2. Characterization and Testing of Samples

The phase of the sample was characterized by an X-ray diffraction analyzer (XRD, DX-2700B, Danton fangyuan Instrument Co., Ltd., Wenzhou, China). Samples were characterized by scanning electron microscopy (SEM, KYKY-EM3900M, Shanghai Zhouyi Instrument Equipment Co., Ltd., Shanghai, China). The pH in the solution was determined by the pH meter (pHS-3C, Shanghai Zhiguang Instrument Co., Ltd., Shanghai, China). The concentration of methyl orange solution was determined by a UV-Visible spectrophotometer (UV 5100B, Shanghai Metash Instrument Co., Ltd., Shanghai, China).

### 2.3. Synthesis of Cu$_2$O

Cu$_2$O was prepared by chemical coprecipitation [22]. The reaction involved is shown in the reaction Equations (1) and (2). First, 120 mL of 0.5 mol/L Cu(NO$_3$)$_2$ solution and 240 mL of 1.5 mol/L NaOH solution were mixed in a beaker and stirred by magnetic force for 30 min. Then, 300 mL of 0.1 mol/L ascorbic acid solution was slowly added to the prepared mixed liquid drop by drop and again stirred for 30 min. Then, the brick red precipitation was formed. After centrifugation, the remaining solid was cleaned several

times by ethanol and deionized water and then dried in a vacuum at 60 °C for 8 h to obtain a brick-red powder.

$$Cu(NO_3)_2 \cdot 3H_2O + 2NaOH \rightarrow Cu(OH)_2 \downarrow + 2NaNO_3 + 3H_2O \tag{1}$$

$$2Cu(OH)_2 + C_6H_8O_6 \rightarrow Cu_2O + C_6H_6O_6 + 3H_2O \tag{2}$$

*2.4. Preparation of Porous Floating Ceramsite Loaded with Cu$_2$O*

Hydrofluoric acid can react with silica originated from floating ceramsite, which can be used to treat floating ceramsite and make them loose and porous on their surface. The reaction process is shown in the chemical Equation (3). Then, the ceramsite was washed to neutral by water.

$$SiO_2 + 4HF \rightarrow SiF_4 + 2H_2O \tag{3}$$

We weighed 100 g of floating ceramsite and put it in a plastic cup. Then, we added the proper amount of distilled water and 100 mL of HF, which caused the porosity of the ceramsite surface to increase. Then, the treated ceramsite was dried at 100 °C and preserved. A certain amount of dried ceramsite was put into the beaker. After that Cu$_2$O was combined with a small amount of anhydrous ethanol. They were stirred slowly for about 0.5 h and dried in a vacuum drying oven for about 8 h. Because Cu$_2$O is easily oxidized to copper oxide slowly in humid air, the formation of CuO can be prevented by isolating the air source. Anhydrous ethanol could be a great choice. Finally, the last product was marked as PFCC.

*2.5. Evaluating the Degradation Effect of MO*

The degradation rate ($\eta$) of MO can be expressed by Formula (1).

$$\eta = \frac{C_0 - C}{C_0} \times 100\% \tag{4}$$

where $C$ and $C_0$ represent the concentration of MO before and after degradation, respectively (mg/L).

The expression of photocatalytic kinetics can often be expressed by the Langmuir–Hinshelwood model, as shown in Formula (2) [23]. When these conditions are satisfied at the same time, which is $KC \ll 1$, $C = C_0$ at $t = 0$, $C = C$ at $t = t$, Equation (5) can be simplified.

$$r = -\frac{dc}{dt} = \frac{k_r KC}{1 + KC} \Rightarrow r = -\frac{dc}{dt} = k_r KC (KC << 1) \Rightarrow -\ln\left(\frac{C}{C_0}\right) = k_1 t \tag{5}$$

where $r$ is the rate of reaction (mg/(L·min)), $k_r$ is the limiting rate constant of reaction at maximum coverage under the given experimental conditions (mg/(L·min)), $K$ is the equilibrium constant of the reactant (L/mg), and $k_1$ is the apparent first-order rate constant ($k_1 = k_r K$, min$^{-1}$) [24].

The degradation amount ($D_a$) of MO by the prepared loaded photocatalyst can be expressed by Equation (3). Moreover, the Langmuir and Freundlich models [25] are used to fit the results of isothermal degradation of dyes as shown in Formulas (4) and (5).

$$D_a = \frac{(C_0 - C)V}{m} \tag{6}$$

$$\frac{C_e}{D_e} = \frac{C_e}{D_m} + \frac{1}{D_m K_L} \tag{7}$$

$$\ln D_e = \frac{1}{n} \ln C_e + \ln K_F \tag{8}$$

where $V$ is the volume of MO solution (L), $m$ is the mass of photocatalyst (mg), $C_e$ is the concentration at which MO is degraded to equilibrium (mg/L), $D_e$ is the degradation

amount of MO when it is degraded to equilibrium (mg/g), $D_m$ is the maximum degradation amount of MO (mg/g), and $n$ is the affinity of two substances. In addition, $K_L$ and $K_F$ are the thermodynamic constants of the Langmuir and Freundlich models, respectively.

### 2.6. Experimental Procedure for Photocatalytic Degradation of MO

Firstly, the loading rates of $Cu_2O$ were considered. PFCC with loading rates of 0%, 2%, 4%, 6%, 8% and 10% was dried for sparing. The photocatalytic degradation experiments were carried out in an MO solution of 20 mg/L. Two grams of PFCC was added to 200 mL MO solution. The pH value of the reaction solution was 4. After 0.5 h of magnetic stirring for the dark reaction, an 8 W (230–280 nm) fluorescent lamp was used to simulate illumination for 1 h. The supernatant was taken and filtered with a 0.45 μm membrane for colorimetric determination.

Secondly, we investigated what effect the dosage of PFCC had on the degradation of MO. We selected PFCC with a $Cu_2O$ loading rate of 8% to treat the MO solution. PFCC dosages of 5, 10, 15, 20, and 25 g/L, respectively, were employed. While keeping the other condition constant, the objects to be treated were treated by magnetic stirring for 0.5 h, and then irradiated for 1 h with an 8 W (230–280 nm) fluorescent lamp as a simulated light source. Afterward, the supernatant was filtered and determined.

Thirdly, photocatalytic time is an important factor in the degradation of MO. With a cuprous oxide of 8% and a dosage of 20 g/L, the photocatalytic time was set to 0.5, 1, 1.5, 2, 2.5 h, respectively. In addition, the dark reaction was carried out simultaneously for 0.5 h. After the reactions had continued for a period of time, the supernatant was filtered and determined.

Fourthly, we considered the pH value of methyl orange solution. Ten beakers containing MO solution were divided into two groups, one for photocatalytic experiments and the other for dark reaction experiments. The pH values of the solutions in each group were adjusted to 2, 4, 6, 8, and 10, respectively. The photocatalytic reaction time was 2 h, and the dark reaction time was 0.5 h. The other experimental conditions remained the same as those described above.

Fifthly, we also investigated the influence of the initial concentration of the MO solution on the photocatalytic effect. The initial concentration of the MO solution was adjusted to 10, 20, 30, 40, 50 mg/L, respectively. Here, the pH value of the MO solution was set to 8. The other experimental conditions were the same as those used in the fourth point.

## 3. Results and Discussion

### 3.1. Characterization of Various Products

The XRD patterns of loaded ceramsite, raw ceramsite, and cuprous oxide are shown in Figure 1. The diffraction peaks of cuprous oxide were 36.40°, 42.28°, 61.36°, and 29.54°, which conform to the known PDF #78-2076 card. In addition, the corresponding crystal plane indices were (111), (200), (220), and (110), respectively. In addition, the crystal belonged to the cubic system. The raw ceramsite had obvious diffraction peaks of silicon oxide and hercynite, which were in good agreement with the PDF #77-1060 and #89-1679 cards, respectively [26]. The loaded ceramsite also had many of the same diffraction peaks as the raw ceramsite. However, unlike raw ceramsite, the loaded ceramsite had more redundant and stronger peaks. Although both of them had partial diffraction peaks aligned with those of cuprous oxide, this was more obvious in the loaded ceramsite, especially at 36.40° and 42.28°. It might be that the low loading of cuprous oxide on the surface of the loaded ceramsite resulted in the cuprous oxide having indistinct peaks.

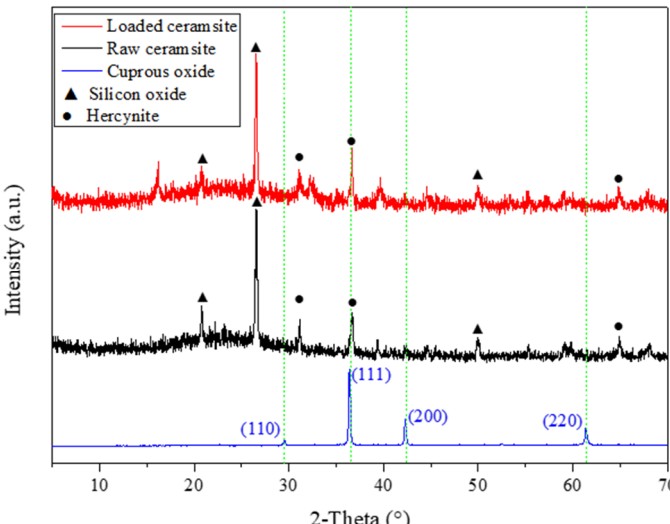

**Figure 1.** XRD pattern of loaded ceramsite, raw ceramsite, and cuprous oxide.

The SEM images for the above three materials and PFCC after degradation of MO are shown in Figure 2. The $Cu_2O$ crystal in Figure 2a had no other impurities, but tended to agglomerate. Perhaps there is a lack of dispersants. The raw ceramsite in Figure 2b can be observed to have many holes, and almost no other impurities in and around the holes. The porosity is about 60–65%, and the pore size is 5–300 μm. The PFCC in Figure 2c can be observed to have a large number of crystalline grains in the hole and a small number of crystalline grains and impurities around the hole. This indicates that $Cu_2O$ was successfully loaded on the ceramsite. In Figure 2d, the PFCC after degradation of MO can be observed to still have a mass of holes. Furthermore, there are only a few impurities in the larger holes. This indicates that after the photocatalytic reaction of the PFCC and MO, the PFCC does not change, and the cuprous oxide maintains a good adhesion property to the suspended porous ceramic support.

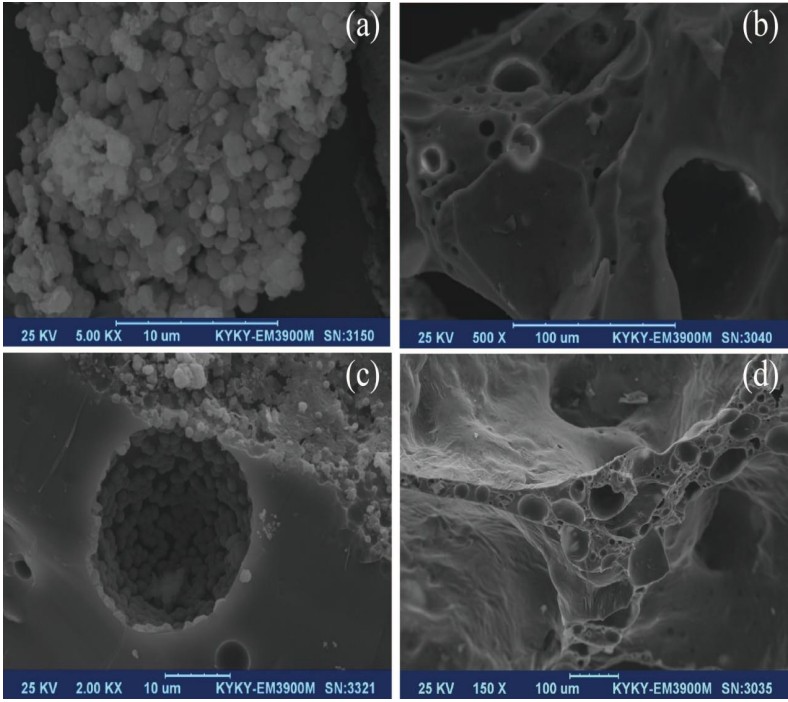

**Figure 2.** SEM images of: (**a**) cuprous oxide; (**b**) raw ceramsite; (**c**) PFCC; (**d**) PFCC after degradation of MO.

To further judge whether $Cu_2O$ was successfully loaded on PFC, the surface elements of the ceramsite were analyzed by EDS (energy dispersive spectrometry), as shown in Figure 3. In Figure 3a,b, the surface of PFC is filled with snowflake-like substances, high in oxygen and copper elements. In addition, the snowflake-like substances are very densely packed. In Figure 3c,d, there are many rod-like substances and small holes with different depths. Elemental analysis showed that the contents of oxygen, aluminum, and iron were very high, indicating that hercynite was indeed present. In Figure 3e,f, rod-like substances and snowflake-like substances coexist, whereby the snowflake-like substances are obviously dispersed on the surface of the rod-like substances. In addition, high copper content was again found in the elemental analysis. In addition, compared with the PFCC, the copper content decreased significantly in the PFCC after degradation of MO. Therefore, it can be concluded that $Cu_2O$ was successfully loaded on ceramsite. Consequently, it can also be inferred that the photo corrosion phenomenon of $Cu_2O$ took place, or that the phenomenon of partial abscission occurred in the reaction process.

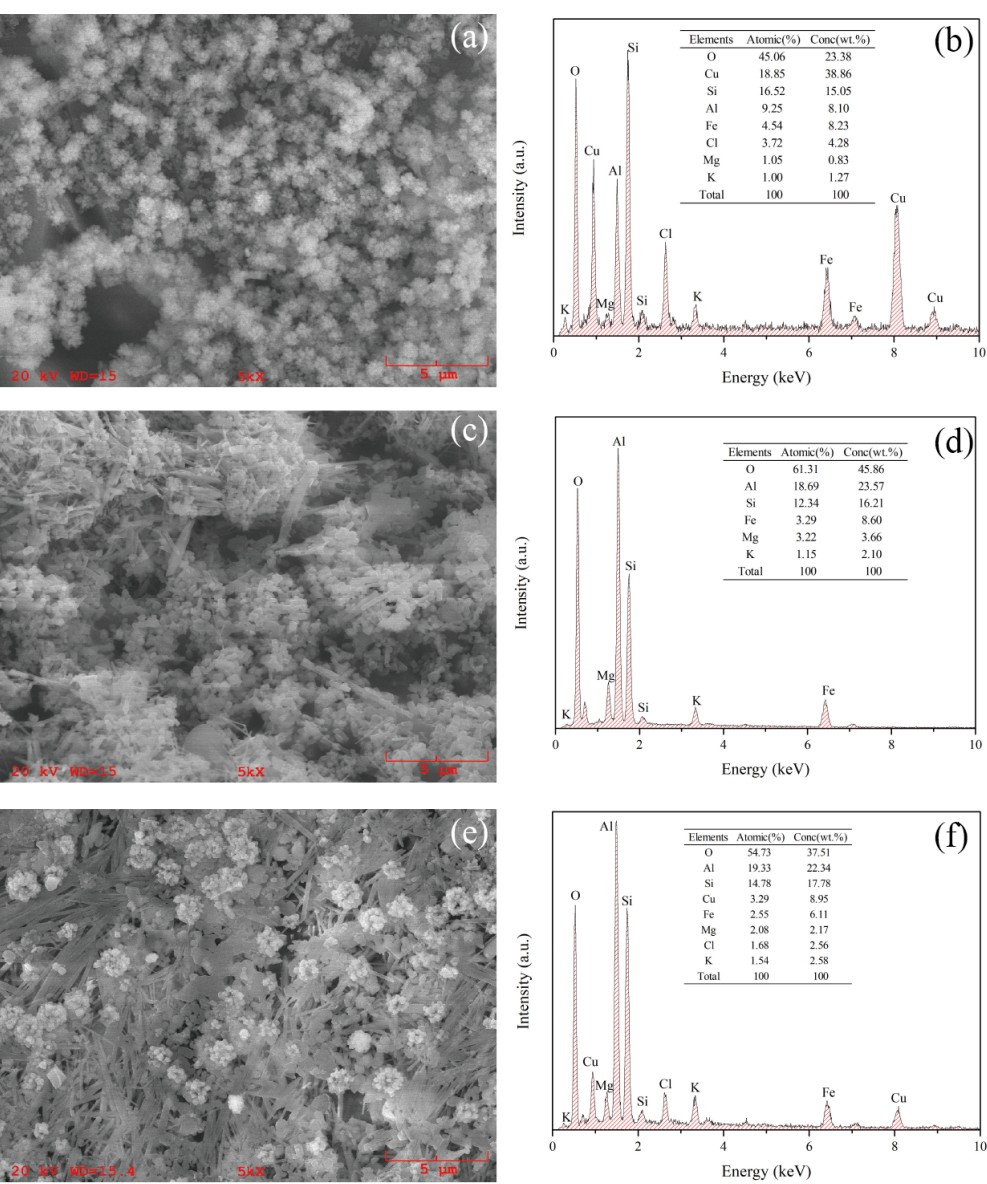

**Figure 3.** The images of SEM and EDS: (**a**,**b**) are PFCC; (**c**,**d**) are PFC; (**e**,**f**) are PFCC after degradation of MO.

### 3.2. Effect of Cuprous Oxide Loading

In Figure 4, when increasing the loading rate of cuprous oxide, both the degradation rate and the amount of degradation increased in the dark reaction and the photocatalytic reaction. In the dark reaction, with the augment of the loading rate of $Cu_2O$, the degradation rate and degradation amount increased slowly in a wave-like manner. In addition, the maximum degradation rate and the maximum degradation amount were 13.12% and 0.26 mg/g, respectively. Unlike in the dark reaction, when the $Cu_2O$ loading rate was increased in the photocatalytic reaction, the degradation rate and degradation amount exhibited a Γ-type growth mode. In addition, the maximum degradation rate and the maximum degradation amount were 41.76% and 0.84 mg/g, respectively. When the loading rate of $Cu_2O$ was 8%, the degradation rate and the degradation amount of MO remained almost unchanged in the photocatalytic reaction. The reason for this might be that the accumulation of cuprous oxide particles hindered the entry of light. Furthermore, the degradation rate and the degradation amount of MO in the photocatalytic reaction were always higher than those in the dark reaction. Therefore, a $Cu_2O$ loading rate of 8% was chosen as the optimal condition for subsequent experiments.

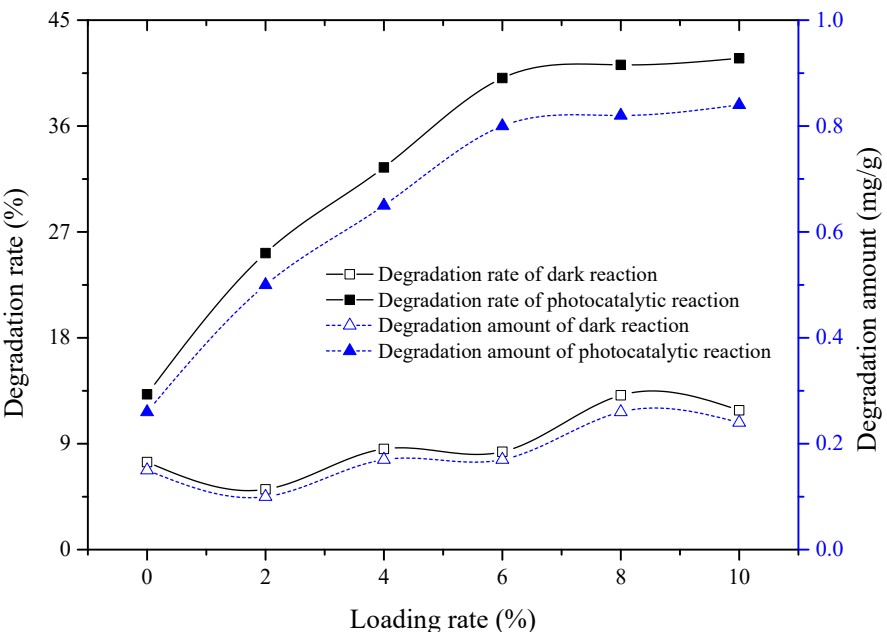

**Figure 4.** The loading rate of $Cu_2O$ had an effect on photocatalysis.

### 3.3. Effect of PFCC Dosage

As shown in Figure 5, the degradation rate of MO increased with increasing dosage of PFCC in both the dark and photocatalytic reactions. In the dark reaction, the amount of MO degraded remained basically unchanged at about 0.15 mg/g with increasing PFCC dosage. In addition, the highest degradation rate of MO was 20.96% in the dark reaction. However, the highest degradation rate and degradation amount of MO reached 63.12% and 0.83 mg/g, respectively, in the photocatalytic reaction, wherein the maximum degradation amount of MO first increased and then decreased with increasing PFCC dosage, and reached its maximum value when the dosage was 10 g/L. In order to achieve a higher degradation rate of MO, a PFCC dosage of 20 g/L was selected for subsequent experiments, as this was the point at which the degradation rate of MO reached equilibrium.

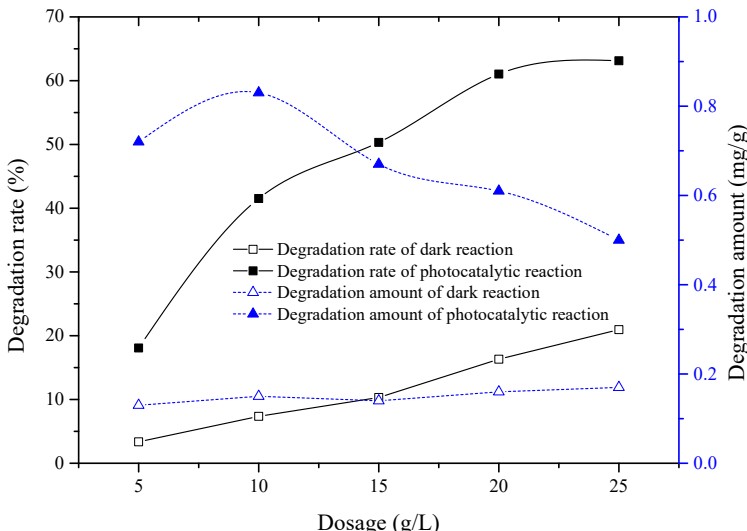

**Figure 5.** The effect of PFCC dosage on photocatalysis.

It can be seen that starting from the dosage of 20 g/L, even if the dosage is increased, the loaded ceramsites pile up and collide with each other during the photocatalytic degradation process, which may affect the scattering of light and may hinder the photocatalytic degradation of methyl orange. When the other conditions remained unchanged, a 5 g/L increase in dosage increases the treatment efficiency by <2%. From the perspective of saving costs and reducing resource waste, we selected a dosage of 20 g/L. Ceramsite was loaded with a load rate of 8% to achieve the best dosage.

### 3.4. Effect of Reaction Time

The degradation rate and the degradation amount of MO increased with the prolongation of photocatalytic reaction time, as shown in Figure 6. Moreover, the maximum degradation rate and the maximum degradation amount of MO were 88.16% and 0.88 mg/g, respectively, at a reaction time of 2 h. However, the degradation rate and amount of MO after 0.5 h of dark reaction were 19.60% and 0.20 mg/g, respectively, values which were lower than those obtained after photocatalysis for the same amount of time. Finally, the photocatalytic reaction time of 2 h was determined to be the optimal photocatalytic reaction time, because this is the point at which the degradation rate and the amount of MO basically reached equilibrium.

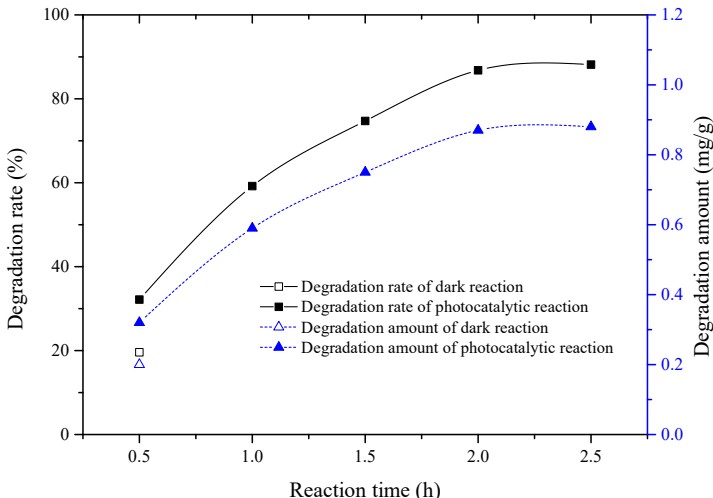

**Figure 6.** The effect of reaction time on the degradation of MO.

### 3.5. Effect of Solution pH

Figure 7 shows that the degradation rate of MO increased slowly in the dark reaction with increasing pH of the solution. In addition, the maximum degradation rate and degradation amount of MO were 21.76% and 0.22 mg/g, respectively. At that moment, the pH of the solution was 8. The maximum degradation rate and amount of MO appeared at pH 8, and were 91.76% and 0.92 mg/g, respectively. This is similar to the findings reported in [27]. That is to say, the degradation rate and the degradation amount of MO were lower in acidic and strongly alkaline conditions than in weakly alkaline conditions. Therefore, the optimum value of pH was found to be 8, and this condition was employed for subsequent experiments.

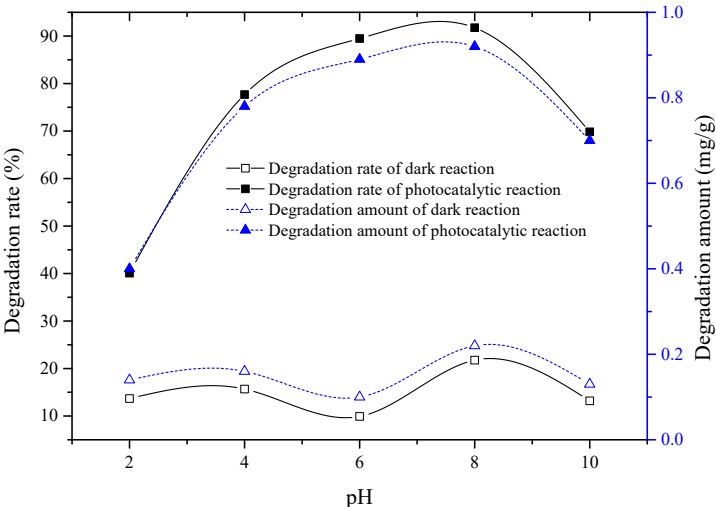

**Figure 7.** Effect of pH value on degradation of MO.

### 3.6. Effect of Initial Concentration of MO

As shown in Figure 8, both the degradation rate and the degradation amount of MO first increased and then decreased with an increasing initial concentration of MO, and their maximum values were 20.69% and 0.31 mg/g, respectively. At that time, the initial concentration of MO was 30 mg/L. Like the dark reaction, the degradation rate of MO first increased and then decreased with increasing initial concentration of MO in the photocatalytic reaction. Its maximum value also appeared with 30 mg/L of MO solution, and was 92.05%. However, unlike the dark reaction, the degradation amount of MO increased almost linearly with increasing initial concentration of MO in the photocatalytic reaction. Therefore, the initial concentration of 30 mg/L of MO was selected as one of the optimum experimental conditions.

It can be seen that when the other conditions were the same, the OH produced by cuprous oxide was the same. With increasing concentration of methyl orange, the photocatalytic degradation efficiency of methyl orange also increased; when the concentration of methyl orange was greater than 30 mg·L$^{-1}$, with increasing initial concentration of methyl orange, the concentration of methyl orange gradually exceeded the maximum value that could be oxidized by OH. In the base orange solution, sufficient OH [28] cannot be formed on the surface of cuprous oxide. Therefore, considering the degradation rate of methyl orange and the cost of industrial processing, the optimum concentration of methyl orange was found to be about 30 mg/L.

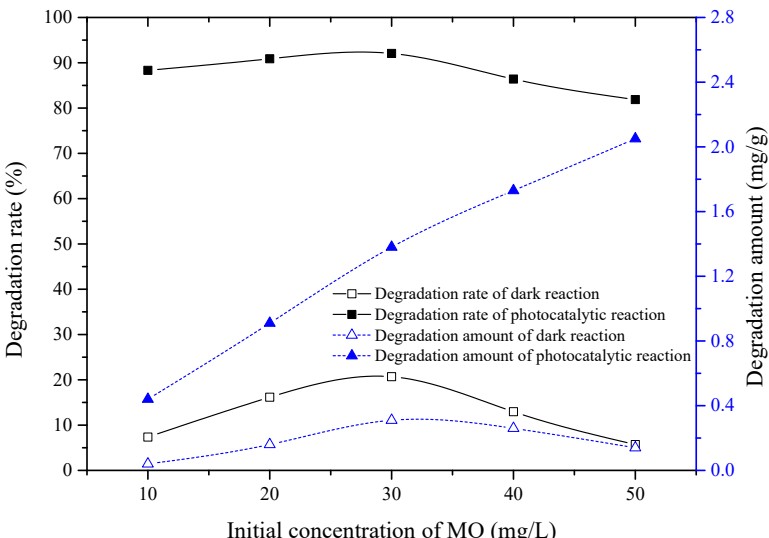

**Figure 8.** Effect of initial concentration of MO on its degradation.

### 3.7. Kinetic and Thermodynamic Analysis

Figure 9 shows that the photocatalytic degradation of MO with time conformed to the Langmuir–Hinshelwood kinetics model. The rate constant $k_1$ of the reaction is 1.186 min$^{-1}$. It can intuitively be seen from Figure 10 that the degradation process of MO in the photocatalytic reaction was more in line with the Freundlich isothermal model. The results of Analysis of Variance (ANOVA) showed that the $p$ values of the Langmuir and Freundlich isothermal models were less than 0.05, at 0.036 and 0.031, respectively. This shows that both models are well able to describe the degradation process of MO, but that it is more in line with the Freundlich isothermal model. In addition, the corresponding parameters were calculated using the linear equation of the adsorption isotherm models. The results are shown in Table 1.

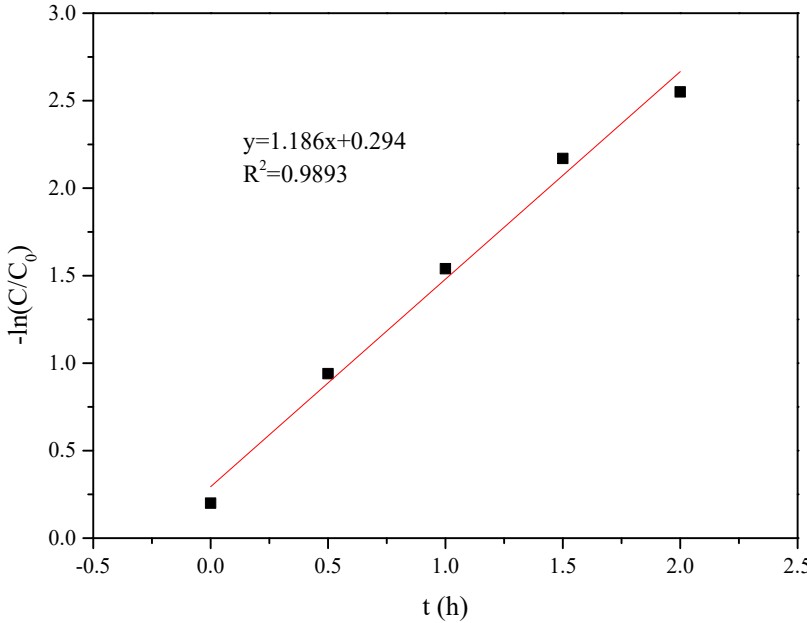

**Figure 9.** Fitting results of Langmuir–Hinshelwood kinetics model.

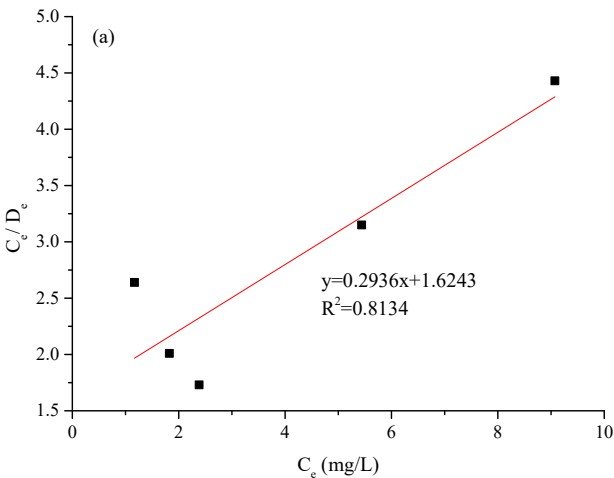

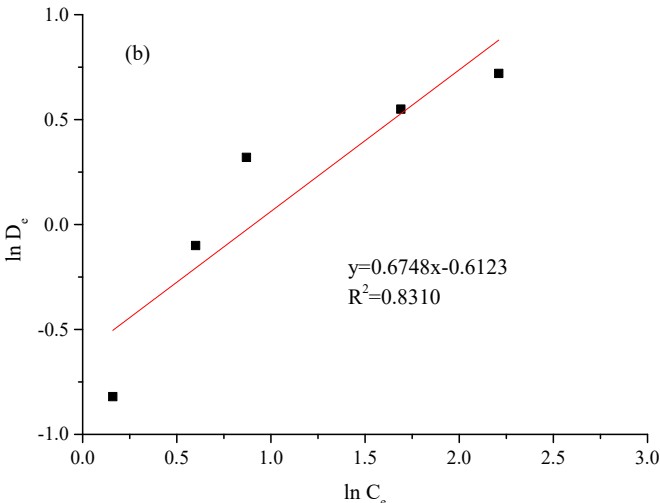

**Figure 10.** Comparison of the Langmuir (**a**) and Freundlich (**b**) isothermal models for MO degradation in the photocatalytic reaction.

**Table 1.** Modeling of isotherms using Langmuir and Freundlich models for MO degradation.

| Isotherm Models | Langmuir | | | Freundlich | | |
|---|---|---|---|---|---|---|
| Parameters | $D_m$ (mg/g) | $K_L$ (L/mg) | $R^2$ | $n$ | $K_F$ (mg/g) | $R^2$ |
| Values | 3.41 | 0.1808 | 0.8134 | 1.48 | 0.5421 | 0.8310 |

Therefore, Table 1 shows that MO molecules were more likely to be adsorbed on the surface of PFCC to form a multi-molecular layer and then catalyzed by photocatalysts. Because Freundlich of $n = 1.48 > 1$, the adsorption of MO on the surface of PFCC belongs to preferential adsorption [29].

### 3.8. Discussion on the Degradation Mechanism of MO

The key experimental steps for the degradation of methyl orange are shown in Figure 11. One of the factors affecting the rate of photocatalytic degradation is the adsorption of substrates on the surface of catalysts. In addition, adsorption is the precondition of photocatalytic reaction [30]. Therefore, MO molecules were first adsorbed by PFC and then adsorbed by $Cu_2O$ powders. After that, they were degraded when irradiated by visible light.

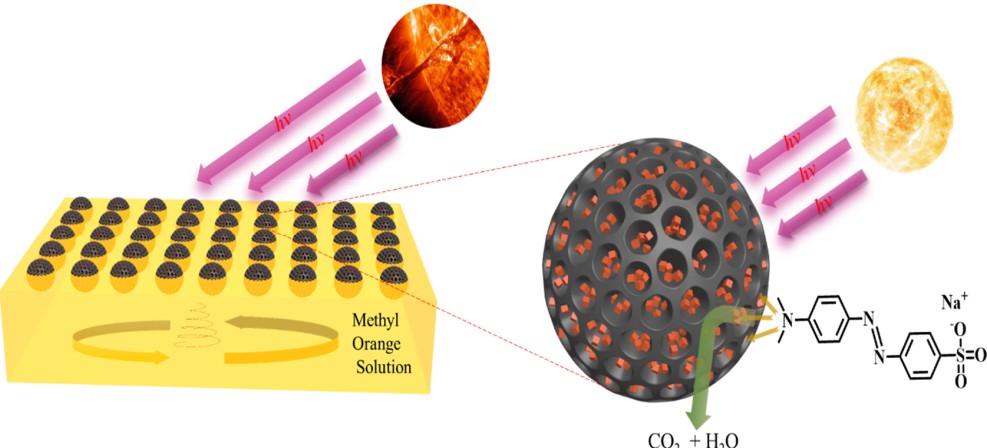

**Figure 11.** Experimental scheme for degradation of MO.

The reason the loading rate of cuprous oxide was not very high, while the degradation rate of MO was high, can be described as follows. The color of PFC is black, which causes it to absorb more colored light. The porous structure of PFC can concentrate light in its pores. The photocatalytic reaction began after $Cu_2O$ absorbed the light. Moreover, the $Cu_2O$ of the (111) crystal plane has better photocatalytic activity than other types [31].

MO mainly exhibits a quinine-type structure in an acidic environment, but an azo-type structure in an alkaline or neutral environment [32–34], as shown in Figure 12. That research found that MO was difficult to degrade in a strongly alkaline environment, but was more easily degraded in an acidic environment. Our results were basically in agreement with this. The difference was that the degradation rate of MO was the highest when the MO solution was weakly alkaline. Therefore, the acid–base conditions of methyl orange will affect its degradation effect.

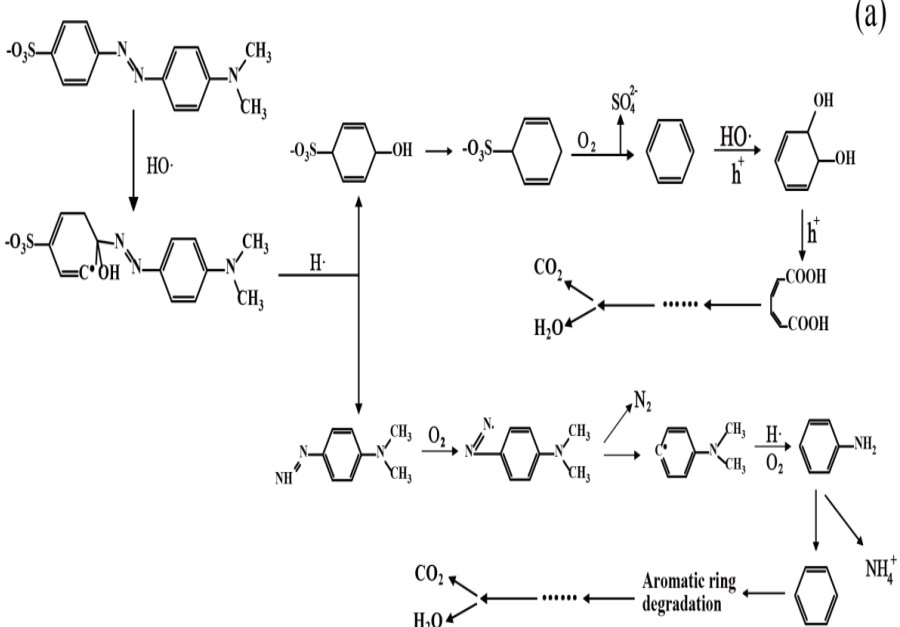

**Figure 12.** *Cont.*

**Figure 12.** The possible processes by which MO is degraded, as elaborated in [22,23,35]: (**a**) azo-type structure in alkaline or neutral solution, (**b**) quinone-type structure in acidic solution, (**c**) degradation path of MO in the presence of hydroxyl radicals.

In the primary process of photocatalysis, photogenerated electrons and holes reacted with oxygen and hydroxyl radicals on the surface of the photocatalyst, respectively. The latter two substances were converted into oxidative superoxide radicals and hydroxyl radicals to participate in photocatalytic reactions [30]. The results are shown in Figure 13.

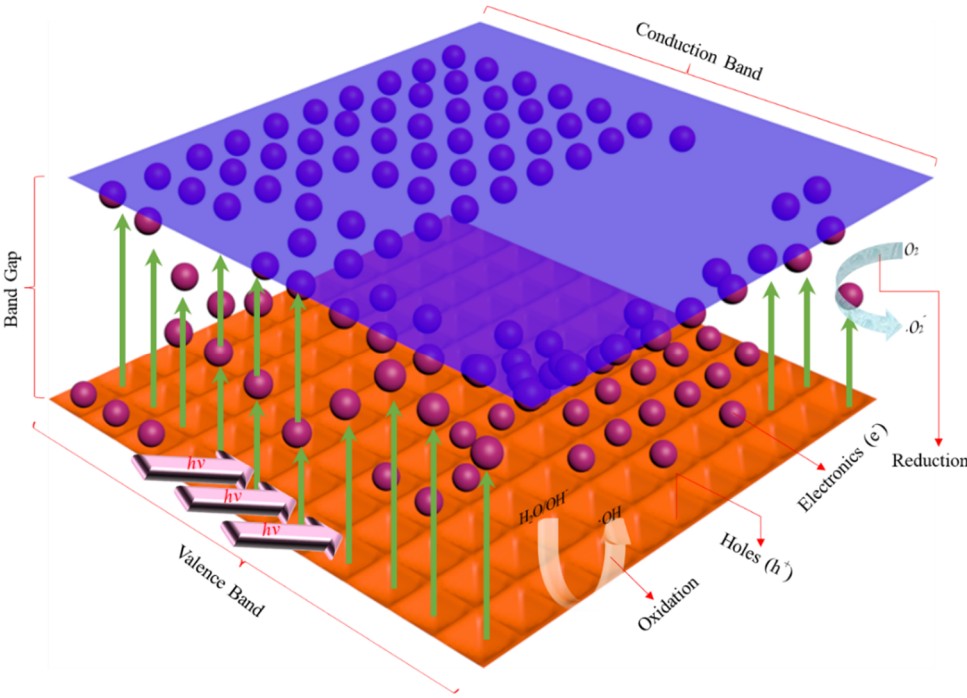

**Figure 13.** Reaction of photocatalyst irradiated by visible light.

Furthermore, the theory of free radicals shows that there are three possible mechanisms for the degradation of MO. These are as follows:

(1)  Degradation mechanism of superoxide free radicals [36]:

$$e^- + O_2 \rightarrow \bullet O_2^- \tag{9}$$

$$O_2^- + H^+ \rightarrow HO_2\bullet \tag{10}$$

$$2HO_2\bullet \rightarrow O_2 + H_2O_2 \tag{11}$$

$$H_2O_2 + \bullet O_2^- \rightarrow \bullet OH + OH^- + O_2 \tag{12}$$

$$H_2O_2 + \bullet OH \rightarrow H_2O + HO_2 \tag{13}$$

$$HO_2\bullet + \bullet OH \rightarrow H_2O + O_2 \tag{14}$$

(2)  Degradation mechanism of hydroxyl radicals [37]:

$$h\nu + Cu_2O \rightarrow h^+ + e^- + Cu_2O \tag{15}$$

$$OH^- + h^+ \rightarrow HO\bullet \tag{16}$$

$$H_2O + h^+ \rightarrow H^+ + HO\bullet \tag{17}$$

$$O_2 + 2H_2O + 2e^- \rightarrow H_2O_2 + 2OH^- \tag{18}$$

$$Cu^{2+} + H_2O_2 \rightarrow Cu^+ + O_2^-\bullet + 2H^+ \tag{19}$$

$$Cu^+ + H_2O_2 \rightarrow Cu^{2+} + OH^- + HO\bullet \tag{20}$$

(3)  Degradation mechanism of holes oxidation [38–40]:

$$HCOCO_2^- + H_2O \leftrightarrow HC(OH)_2CO_2^- \tag{21}$$

$$HC(OH)_2CO_2^- + h^+_{vb} \rightarrow HC(OH)_2CO_2\bullet \tag{22}$$

$$HC(OH)_2CO_2\bullet \rightarrow HC(OH)_2\bullet + CO_2 \tag{23}$$

$$HC(OH)_2\bullet + h^+_{vb} \rightarrow HCO_2^- + 2H^+ \tag{24}$$

Although these three mechanisms perhaps exist at the same time, the degradation mechanism of hydroxyl radicals is the most easily acceptable in the three mechanisms mentioned above. In the process of photocatalytic reaction, adsorption is a very important step. Adsorption is a prerequisite for photocatalytic reaction. Only the pollutant molecules adsorbed on the photocatalyst can be further degraded. The degradation of MO with $Cu_2O$ mainly takes place through photocatalysis, with PFC through adsorption, and with PFCC through the combination of adsorption and photocatalysis. The maximum degradation rate of MO in a dark reaction was only 21.76%. In addition, MO does not degrade itself [41–43]. This indicates that the photocatalytic reaction plays a dominant role, while the adsorption reaction plays an auxiliary role.

Photocatalysts have gone through three generations [43]. The first generation of photocatalysts had a large band gap and a high probability of interfacial charge recombination, which makes it difficult to achieve the degradation of dyes. The second generation of photocatalysts had many advantages, but they were not suitable for industrial-scale applications. The main characteristic of the third generation of photocatalysts is the reclamation of immobilization technology, and they will continue to be developed.

## 4. Conclusions

Porous floating ceramsite loaded with cuprous oxide (PFCC) was successfully prepared. These materials can find suitable application as adsorbents and effective photocatalysts. The obtained materials were characterized by XRD and SEM including EDS.

The optimal photocatalytic conditions for the degradation of methylene orange (MO) were as follows: the loading rate of cuprous oxide was 8%; the dosage of PFCC was 20 g/L; the reaction time of PFCC with MO solution was 2 h; the pH of the MO solution was 8; the initial concentration of MO solution was 30 mg/L.

Furthermore, the maximum degradation rate of MO was 92.05%. The experimental reaction kinetics of photocatalytic degradation fits very well to the Langmuir–Hinshelwood model. The experimental reaction thermodynamics of photocatalytic degradation is in accordance with the Freundlich model. At the macro level, the degradation of MO is most probably due to the synergetic effect of adsorption and photocatalytic oxidation, whereas at the micro level, the degradation of MO occurs via production of a large number of hydroxyl radicals in the photocatalytic process, which continuously oxidize MO, producing $CO_2$ and $H_2O$.

**Author Contributions:** Conceptualization, Y.C.; methodology, Y.C.; software, Z.X.; validation, T.C. and M.Y.; formal analysis, M.Y.; investigation, T.C. and H.Z.; resources, Y.C.; data curation, T.C.; writing—original draft preparation, Z.X. and T.C.; writing—review and editing, Y.C., T.C. and Z.X.; supervision, Y.C.; project administration, Y.C.; funding acquisition, Y.C. All authors have read and agreed to the published version of the manuscript.

**Funding:** This study received strong support from the National Natural Science Foundation of China (No. 51268018) and Jingdezhen Science and Technology Bureau Project (20202GYZD013-20).

**Institutional Review Board Statement:** Not applicable.

**Informed Consent Statement:** Not applicable.

**Data Availability Statement:** Not applicable.

**Acknowledgments:** The authors acknowledge the support provided by National Engineering Research Center for Domestic and Building Ceramics at the Jingdezhen Ceramic Institute with respect to analytical measurements.

**Conflicts of Interest:** The authors declare no conflict of interest.

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
