# Peer review of "Photocatalytic Treatment of Methyl Orange Dye Wastewater by Porous Floating Ceramsite Loaded with Cuprous Oxide"

_coatings, doi:10.3390/coatings12020286_

Round 1

Reviewer 1 Report

Review of the Manuscript ID Coatings-1542800

This work is dedicated to Photocatalytic treatment of methyl orange dye wastewater by

3 porous floating ceramsite loaded with cuprous oxide

I suggest acceptance of the paper after major revision.

Meanwhile I have some questions, notes and recommendations.

  1. English should be improved and Abstract, Introduction, Results and Conclusions should be rewritten.

Abstract

Printing and dyeing wastewater is one of the industrial wastewaters which is difficult to treat. For decomposing dyes from dyestuff wastewater into almost harmless substances in the natural environment, an easily prepared, efficient, and practical composite material is easy to prepare, which is a easy to recycle and regenerate of porous floating ceramsite loaded with cuprous oxide (PFCC), was developed in this study. The PFCCs were prepared and characterized by X-ray diffraction spectroscopy (XRD), scanning electron microscopy (SEM), and energy dispersive spectrometer (EDS). The material was used for photocatalytic degradation of methyl orange (MO) in water. The results showed that the maximum degradation rate of MO was 92.05% when the experimental conditions were cuprous oxide loading rate of 8%, PFCC dosage of 20 g/L, the reaction time of 2 h, pH value of 8, and solution’s initial concentration of 30 mg/L. The degradation processes of MO accorded with the Langmuir-Hinshelwood model in reaction kinetics and Freundlich model in reaction thermodynamics respectively. The degradation mechanism of MO might be considered from  two perspectives. One was the synergistic effect of adsorption and photocatalytic oxidation, and the other was the strong oxidation of hydroxyl radicals produced by photocatalysts.

The text should be corrected as follows:

“It is well known that water treatment of printing and dyeing wastewaters is problematic. For decomposing dyes from dyestuff wastewater and conversion into almost harmless substances for the natural environment, an easily prepared, efficient, practical and easy to regenerate composite material from porous floating ceramsite loaded with cuprous oxide (PFCC). The PFCC samples were prepared and characterized by X-ray diffraction spectroscopy (XRD), scanning electron microscopy (SEM), and energy dispersive spectroscopy(EDS). The material was applied for photocatalytic degradation of methyl orange (MO) in water. The results show that the maximal degradation rate of MO was 92.05 % when the experimental conditions were: cuprous oxide loading rate of 8%, PFCC dosage of 20 g/L, the reaction time of 2 h, pH value of 8, and solution initial concentration of 30 mg/L. The degradation processes of MO fits well with the Langmuir-Hinshelwood model in reaction kinetics, and Freundlich model in reaction thermodynamics, respectively. The degradation mechanism of MO might be considered from two perspectives- one was the synergetic effect of adsorption and photocatalytic oxidation, and the other was the strong oxidation of hydroxyl radicals produced by photocatalysts.”

  1. Introduction, rows 25-30

“China is the world's largest country in the production and consumption of dyes, and the amount of dye and printing and dyeing wastewater pollution broad in scope [1-2]. Dye and printing and dyeing wastewater is one of the most difficult industrial wastewaters. It has the characteristics of large water volume, high chroma, complex chemical composition, and difficult biochemical degradation. The main pollutants in this type of wastewater are BOD, COD, organic toxic substances and chromaticity [3-4].”

The text is not correct. It should be rewritten as follows:

“China is the world's largest country in the production and consumption of dyes, and the amount of dye and printing wastewater, and dyeing wastewater pollution continue to increase [1-2]. Dye and printing wastewater, and dyeing wastewater are among the most problematic industrial wastewaters. Dye wastewaters are emitted in large volume, and they are distinguished by high chroma, complex chemical composition, and they are not suitable for effective biochemical degradation. The main methods for detection of pollutants in this type of wastewaters are BOD (biochemical oxygen demand) analysis, COD (chemical oxygen demand) analysis, determination of organic toxic substances and chromaticity [3-4].”

  1. Introduction, rows 33-35

“Studies have shown that using TiO2, ZnO, CdS, etc. as photocatalysts can effectively degrade the content in wastewater [8]. Organic substances such 35 as dyes.”

Should be replaced with:

“Studies have shown that using TiO2, ZnO, CdS, etc. as photocatalysts can effectively degrade organic substances such as dyes in wastewater [8].”

  1. Introduction, rows 40-41

“The practical value of powders in photocatalytic applications is reduced due to their recovery defects.”

Should be substituted by:

“The practical value of powders in photocatalytic applications is reduced due to their recovery problems.”

  1. Materials and methods

2.4. Preparation of porous floating ceramsite loaded with Cu2O

“Hydrofluoric acid can react with silica originated from floating ceramsite, which can be used to treat floating ceramsite and make them loose and porous on their surface.”

Should be replaced by:

“Hydrofluoric acid can react with silica originated from floating ceramsite, leading to formation of pores in ceramsite.”

6.Conclusions

“We have successfully prepared PFCC. These materials had the functions of adsorption and photocatalysis at the same time. But photocatalysis played a dominant role. The PFCC was characterized by XRD and SEM (including EDS). The optimum photocatalytic conditions for the degradation of MO are as follows. The loading rate of cuprous oxide was 8%. The dosage of PFCC was 20 g/L. The reaction time of PFCC with MO solution was 2 h. The pH of the MO solution was 8. And the initial concentration of MO solution was 30 mg/L. Furthermore, the maximum degradation rate of MO was 92.05%. The reaction kinetics of photocatalytic degradation conformed to Langmuir- Hinshelwood model. The reaction thermodynamics of photocatalytic degradation conformed to the Freundlich model. At the macro level, the degradation of MO might be due to the synergistic effect of adsorption and photocatalytic oxidation; at the micro-level, the degradation of MO might be due to the production of a large number of hydroxyl radicals in the photocatalytic process, which continuously oxidize MO until it became CO2 and H2O.”

Conclusions should be rewritten as follows:

“Porous floating ceramsite loaded with cuprous oxide (PFCC) was successfully prepared. These materials found suitable application as adsorbents and effective photocatalysts. The obtained materials were characterized by XRD and SEM including EDS.

The optimal photocatalytic conditions for the degradation of methylene orange (MO)  are as follows: The loading rate of cuprous oxide was 8 %; The dosage of PFCC was 20 g/L; The reaction time of PFCC with MO solution was 2 h; The pH of the MO solution was 8; The initial concentration of MO solution was 30 mg/L.

Furthermore, the maximal degradation rate of MO was 92.05 %. The experimental reaction kinetics of photocatalytic degradation fits very well to Langmuir- Hinshelwood model. The experimental reaction thermodynamics of photocatalytic degradation is in accordance with Freundlich model. At the macro level, the degradation of MO is most probably due to the synergetic effect of adsorption and photocatalytic oxidation; whereas at the micro-level, the degradation of MO occurs via production of a large number of hydroxyl radicals in the photocatalytic process, which continuously oxidize MO, producing CO2 and H2O.”

  1. Please, if possible, provide BET results for investigated materials.

  1. Authors should correct the references according to

Author Response

Reply to reviewer 1

Review of the Manuscript ID Coatings-1542800

This work is dedicated to Photocatalytic treatment of methyl orange dye wastewater by

3 porous floating ceramsite loaded with cuprous oxide

I suggest acceptance of the paper after major revision.

Meanwhile I have some questions, notes and recommendations.

  1. English should be improved and Abstract, Introduction, Results and Conclusions should be rewritten.

Abstract

Printing and dyeing wastewater is one of the industrial wastewaters which is difficult to treat. For decomposing dyes from dyestuff wastewater into almost harmless substances in the natural environment, an easily prepared, efficient, and practical composite material is easy to prepare, which is a easy to recycle and regenerate of porous floating ceramsite loaded with cuprous oxide (PFCC), was developed in this study. The PFCCs were prepared and characterized by X-ray diffraction spectroscopy (XRD), scanning electron microscopy (SEM), and energy dispersive spectrometer (EDS). The material was used for photocatalytic degradation of methyl orange (MO) in water. The results showed that the maximum degradation rate of MO was 92.05% when the experimental conditions were cuprous oxide loading rate of 8%, PFCC dosage of 20 g/L, the reaction time of 2 h, pH value of 8, and solution’s initial concentration of 30 mg/L. The degradation processes of MO accorded with the Langmuir-Hinshelwood model in reaction kinetics and Freundlich model in reaction thermodynamics respectively. The degradation mechanism of MO might be considered from  two perspectives. One was the synergistic effect of adsorption and photocatalytic oxidation, and the other was the strong oxidation of hydroxyl radicals produced by photocatalysts.

The text should be corrected as follows:

“It is well known that water treatment of printing and dyeing wastewaters is problematic. For decomposing dyes from dyestuff wastewater and conversion into almost harmless substances for the natural environment, an easily prepared, efficient, practical and easy to regenerate composite material from porous floating ceramsite loaded with cuprous oxide (PFCC). The PFCC samples were prepared and characterized by X-ray diffraction spectroscopy (XRD), scanning electron microscopy (SEM), and energy dispersive spectroscopy(EDS). The material was applied for photocatalytic degradation of methyl orange (MO) in water. The results show that the maximal degradation rate of MO was 92.05 % when the experimental conditions were: cuprous oxide loading rate of 8%, PFCC dosage of 20 g/L, the reaction time of 2 h, pH value of 8, and solution initial concentration of 30 mg/L. The degradation processes of MO fits well with the Langmuir-Hinshelwood model in reaction kinetics, and Freundlich model in reaction thermodynamics, respectively. The degradation mechanism of MO might be considered from two perspectives- one was the synergetic effect of adsorption and photocatalytic oxidation, and the other was the strong oxidation of hydroxyl radicals produced by photocatalysts.”

I have corrected in yellow.

  1. Introduction, rows 25-30

“China is the world's largest country in the production and consumption of dyes, and the amount of dye and printing and dyeing wastewater pollution broad in scope [1-2]. Dye and printing and dyeing wastewater is one of the most difficult industrial wastewaters. It has the characteristics of large water volume, high chroma, complex chemical composition, and difficult biochemical degradation. The main pollutants in this type of wastewater are BOD, COD, organic toxic substances and chromaticity [3-4].”

The text is not correct. It should be rewritten as follows:

“China is the world's largest country in the production and consumption of dyes, and the amount of dye and printing wastewater, and dyeing wastewater pollution continue to increase [1-2]. Dye and printing wastewater, and dyeing wastewater are among the most problematic industrial wastewaters. Dye wastewaters are emitted in large volume, and they are distinguished by high chroma, complex chemical composition, and they are not suitable for effective biochemical degradation. The main methods for detection of pollutants in this type of wastewaters are BOD (biochemical oxygen demand) analysis, COD (chemical oxygen demand) analysis, determination of organic toxic substances and chromaticity [3-4].”

I have rewritten in yellow.

  1. Introduction, rows 33-35

“Studies have shown that using TiO2, ZnO, CdS, etc. as photocatalysts can effectively degrade the content in wastewater [8]. Organic substances such 35 as dyes.”

Should be replaced with:

“Studies have shown that using TiO2, ZnO, CdS, etc. as photocatalysts can effectively degrade organic substances such as dyes in wastewater [8].”

I have replaced in yellow.

  1. Introduction, rows 40-41

“The practical value of powders in photocatalytic applications is reduced due to their recovery defects.”

Should be substituted by:

“The practical value of powders in photocatalytic applications is reduced due to their recovery problems.”

I have substituted in yellow.

  1. Materials and methods

2.4. Preparation of porous floating ceramsite loaded with Cu2O

“Hydrofluoric acid can react with silica originated from floating ceramsite, which can be used to treat floating ceramsite and make them loose and porous on their surface.”

Should be replaced by:

“Hydrofluoric acid can react with silica originated from floating ceramsite, leading to formation of pores in ceramsite.”

I have replaced in yellow.

6.Conclusions

“We have successfully prepared PFCC. These materials had the functions of adsorption and photocatalysis at the same time. But photocatalysis played a dominant role. The PFCC was characterized by XRD and SEM (including EDS). The optimum photocatalytic conditions for the degradation of MO are as follows. The loading rate of cuprous oxide was 8%. The dosage of PFCC was 20 g/L. The reaction time of PFCC with MO solution was 2 h. The pH of the MO solution was 8. And the initial concentration of MO solution was 30 mg/L. Furthermore, the maximum degradation rate of MO was 92.05%. The reaction kinetics of photocatalytic degradation conformed to Langmuir- Hinshelwood model. The reaction thermodynamics of photocatalytic degradation conformed to the Freundlich model. At the macro level, the degradation of MO might be due to the synergistic effect of adsorption and photocatalytic oxidation; at the micro-level, the degradation of MO might be due to the production of a large number of hydroxyl radicals in the photocatalytic process, which continuously oxidize MO until it became CO2 and H2O.”

Conclusions should be rewritten as follows:

“Porous floating ceramsite loaded with cuprous oxide (PFCC) was successfully prepared. These materials found suitable application as adsorbents and effective photocatalysts. The obtained materials were characterized by XRD and SEM including EDS.

The optimal photocatalytic conditions for the degradation of methylene orange (MO)  are as follows: The loading rate of cuprous oxide was 8 %; The dosage of PFCC was 20 g/L; The reaction time of PFCC with MO solution was 2 h; The pH of the MO solution was 8; The initial concentration of MO solution was 30 mg/L.

Furthermore, the maximal degradation rate of MO was 92.05 %. The experimental reaction kinetics of photocatalytic degradation fits very well to Langmuir- Hinshelwood model. The experimental reaction thermodynamics of photocatalytic degradation is in accordance with Freundlich model. At the macro level, the degradation of MO is most probably due to the synergetic effect of adsorption and photocatalytic oxidation; whereas at the micro-level, the degradation of MO occurs via production of a large number of hydroxyl radicals in the photocatalytic process, which continuously oxidize MO, producing CO2 and H2O.”

I have rewritten in yellow.

  1. Please, if possible, provide BET results for investigated materials.

Sorry,I have not provided BET results for investigated materials because I forgot to measure it.

  1. Authors should correct the references according to

I have corrected the references in yellow.

Reviewer 2 Report

Reviewer comments_ coatings-1542800

The authors of this manuscript discuss on photocatalytic treatment of methyl orange dye wastewater by porous floating ceramsite loaded with cuprous oxide. Results reported in this manuscript deserve to be known by other researchers. But before the publication several questions should be illustrated more clearly to make the manuscript more readable and meaningful to readers. Detail comments are as follows:

  1. The language and structure of this manuscript needs to be improved as it is quite hard for readers to understand the key highlights of this manuscript. Abstract and conclusion need to be revised to make readers understand better.
  2. Revise sentence in line 27 “Dye and printing and dyeing wastewater is one of the most difficult industrial wastewater”. Dye and printing, and dyeing wastewater?. What do you mean by difficult in this sentence?
  3. Write the full meaning of BOD, COD in the introduction part
  4. Revise word “photo catalytic” to “photocatalytic”
  5. Revise sentence 164 “ The diffraction peaks of cuprous oxide were 36.40°, 42.28°, 61.36°, and 29.54°, which conform to known PDF #78-2076 card”. What does it mean? Do you need to add this as reference for peak 2θ angle? Can you add reference from previous literature as the reference as well?
  6. Can you clarify on this statement in line 185 “The raw ceramsite in Fig. 2 can be observed to have many holes and almost no other impurities in and around the holes”.How many holes? how big is the hole
  7. In Fig. 2 (d), the PFCC after degradation of MO can be observed to still have a mass of holes. And there are only a few impurities in the larger holes. Kindly support this statement with reason.
  8. What is the wavelength or intensity of the 8 W fluorescent lamp used?
  9. It is suggested to add error bars in all data presented in the Figures to validate the results.
  10. Add support statements to all plausible reasons or discussion presented.
  11. Kindly explain further on this sentence (line 339)“ Although these three mechanisms perhaps exist at the same time, the degradation mechanism of hydroxyl radicals is the most easily acceptable in the three mechanisms mentioned above.”. How about adsorption process? Which process come first? Adsorption or photodegradation although the synergistic effect of adsorption and photocatalytic oxidation could takes place during the experiment.
  12. How about the anionic properties of MO, does it has an interaction with surface properties if PFCC?
  13. It is also suggested that authors to provide data of zeta potential of PFCC to study the interaction of materials with MO.
  14. Overall, the language and quality of manuscript require major improvement.

Author Response

Reply to  Reviewer 2

Reviewer comments_ coatings-1542800

The authors of this manuscript discuss on photocatalytic treatment of methyl orange dye wastewater by porous floating ceramsite loaded with cuprous oxide. Results reported in this manuscript deserve to be known by other researchers. But before the publication several questions should be illustrated more clearly to make the manuscript more readable and meaningful to readers. Detail comments are as follows:

  1. The language and structure of this manuscript needs to be improved as it is quite hard for readers to understand the key highlights of this manuscript. Abstract and conclusion need to be revised to make readers understand better.

I have improved this manuscript in yellow.

  1. Revise sentence in line 27 “Dye and printing and dyeing wastewater is one of the most difficult industrial wastewater”. Dye and printing, and dyeing wastewater?. What do you mean by difficult in this sentence?

Dye and printing wastewater, and dyeing wastewater are among the most problematic industrial wastewaters. I have corrected in line 27 in yellow.

  1. Write the full meaning of BOD, COD in the introduction part

I have written BOD (biochemical oxygen demand) analysis, COD (chemical oxygen demand) in the introduction part.

  1. Revise word “photo catalytic” to “photocatalytic”

I have revised word “photo catalytic” to “photocatalytic”

  1. Revise sentence 164 “ The diffraction peaks of cuprous oxide were 36.40°, 42.28°, 61.36°, and 29.54°, which conform to known PDF #78-2076 card”. What does it mean? Do you need to add this as reference for peak 2θ angle? Can you add reference from previous literature as the reference as well?

The reference is Zhang X, Song J, Jiao J, et al. Preparation and photocatalytic activity of cuprous oxides[J]. Solid State Sciences, 2010, 12(7): 1215-1219.

  1. Can you clarify on this statement in line 185 “The raw ceramsite in Fig. 2 can be observed to have many holes and almost no other impurities in and around the holes”.How many holes? how big is the hole

The porosity is about 60-65%, and the pore size is 5-300μm.

  1. In Fig. 2 (d), the PFCC after degradation of MO can be observed to still have a mass of holes. And there are only a few impurities in the larger holes. Kindly support this statement with reason.

This indicates that after the photocatalytic reaction of the PFCC and MO, the PFCC does not change, and the cuprous oxide maintains a good adhesion property to the suspended porous ceramic support.

  1. What is the wavelength or intensity of the 8 W fluorescent lamp used?

The wavelength or intensity of the 8 W fluorescent lamp is 230-280nm

  1. It is suggested to add error bars in all data presented in the Figures to validate the results.

I will not do such a picture now, and I will learn it in the future.

  1. Add support statements to all plausible reasons or discussion presented.

I have corrected in this manuscript

  1. Kindly explain further on this sentence (line 339)“ Although these three mechanisms perhaps exist at the same time, the degradation mechanism of hydroxyl radicals is the most easily acceptable in the three mechanisms mentioned above.”. How about adsorption process? Which process come first? Adsorption or photodegradation although the synergistic effect of adsorption and photocatalytic oxidation could takes place during the experiment.

In the process of photocatalytic reaction, adsorption is a very important step. Adsorption is a prerequisite for photocatalytic reaction. Only the pollutant molecules adsorbed on the photocatalyst can be further degraded.

  1. How about the anionic properties of MO, does it has an interaction with surface properties if PFCC?

This requires further in-depth research.

  1. It is also suggested that authors to provide data of zeta potential of PFCC to study the interaction of materials with MO.

The zeta potential was not measured due to the lack of a zeta potentiometer and forgot to measure it.

  1. Overall, the language and quality of manuscript require major improvement.

I have improved language and quality of manuscript  

Reviewer 3 Report

The study synthesized porous floating ceramsite loaded with cuprous oxide, and investigate their photocatalytic activities in degradation methyl orange in waters. Overall, the work is interesting for the readers of Coatings. However, issues should be addressed before it is considered for publication.

Please, go through the specific comments, questions, and typos below:

  1. Please re-write to improve the following sentence: “For decomposing dyes from dyestuff wastewater into almost harmless substances in the natural environment, an easily prepared, efficient, and practical composite material is easy to prepare, which is a easy to recycle and regenerate of porous floating ceramsite loaded with cuprous oxide (PFCC), was developed in this study.”
  2. Line 34-35, no verb in the senence: “Organic substances such as dyes”
  3. Two “dye” in the noun phrase of “dye and printing and dyeing wastewater”
  4. Line 112, 131 photo catalytic -> photocatalytic
  5. Lines 244 – 245, it needs to write the reaction time for 88.16% and 0.88 mg/g.
  6. The two sentences in lines 255 – 257 should be combined.
  7. The authors wrote: “Because of 1/2<n=1.48<2, MO was not difficult to adsorb on the surface of PFCC”. Please explain?
  8. Please explain the opposite variation tendency between MO photocatalytic degradation rate curve and MO photocatalytic degradation amount curve in Figure 5 and Figure 8.
  9. I strongly recommend to improve the English of the manuscript.
  10. Increase the font size and numbers in x- and y- axis in Figures 9, 10.
  11. Further writing about three generations of photocatalysts is of interest.
  12. What is your message with the sentence: “So our work was to prepare the third generation photocatalyst”.
  13. The authors should compare the photocatalytic performance of PFCC in this study with the other well-known photocatalysts in degradation dyes.
  14. For the optimizing procedure, why you chose to the optimize order presented in the manuscript (i.e., Cu2O loading, dosage, reaction time, pH, MO initial concentration.
  15. Is there heating effect during the photocatalytic reaction under irradiation from 8W fluorescent lamp?   
  16. The absorption spectra in UV-VIS-NIR IR range of as prepared Cu2O, PFC, PFCC samples are of interests to elucidate the roles of each material in the degradation of MO.

Typos

Line 298,reactio”

Author Response

Reply to  Reviewer 3

Comments and Suggestions for Authors

The study synthesized porous floating ceramsite loaded with cuprous oxide, and investigate their photocatalytic activities in degradation methyl orange in waters. Overall, the work is interesting for the readers of Coatings. However, issues should be addressed before it is considered for publication.

Please, go through the specific comments, questions, and typos below:

  1. Please re-write to improve the following sentence: “For decomposing dyes from dyestuff wastewater into almost harmless substances in the natural environment, an easily prepared, efficient, and practical composite material is easy to prepare, which is a easy to recycle and regenerate of porous floating ceramsite loaded with cuprous oxide (PFCC), was developed in this study.”

For decomposing dyes from dyestuff wastewater and conversion into almost harmless substances for the natural environment, an easily prepared, efficient, practical and easy to regenerate composite material from porous floating ceramsite loaded with cuprous oxide (PFCC).

  1. Line 34-35, no verb in the senence: “Organic substances such as dyes”

Studies have shown that using TiO2, ZnO, CdS, etc. as photocatalysts can effectively degrade organic substances such as dyes in wastewater [8]

  1. Two “dye” in the noun phrase of “dye and printing and dyeing wastewater”

Dye and printing wastewater

  1. Line 112, 131 photo catalytic -> photocatalytic

Yes

  1. Lines 244 – 245, it needs to write the reaction time for 88.16% and 0.88 mg/g.

at reaction time of 2 h

  1. The two sentences in lines 255 – 257 should be combined.

But in the photocatalytic reaction, the degradation rate and amount of MO increased first and then decreased.have been deleted.

  1. The authors wrote: “Because of 1/2<n=1.48<2, MO was not difficult to adsorb on the surface of PFCC”. Please explain?

  1. Please explain the opposite variation tendency between MO photocatalytic degradation rate curve and MO photocatalytic degradation amount curve in Figure 5 and Figure 8.

It can be analyzed in Figure 5 that starting from the dosage of 20g/L, even if the dosage is in-creased, during the photocatalytic degradation process, the loaded ceramsites pile up and collide with each other, which may affect the scattering of light and may hinder the photocatalytic degradation of methyl orange; other Under the condition that the conditions remain unchanged, increasing the dosage of 5g/L increases the treatment efficiency by <2%. From the perspective of saving costs and reducing waste of re-sources, the selected dosage is 20g/L. Loaded ceramsite with a load rate of 8% for the best dosage.

It can be analyzed in Fig.8 that under the same other conditions, the OH produced by cuprous oxide is the same. As the concentration of methyl orange increases, the photocatalytic degradation efficiency of methyl orange also increases; when the concentration of methyl orange is greater than 30mg/ When L, with the increase of the initial concentration of methyl orange, one is that the concentration of methyl orange gradually exceeds the maximum value that can be oxidized by OH; In the base orange solution, sufficient OH[27] cannot be formed on the surface of cuprous oxide. Therefore, considering the degradation rate of methyl orange and the cost of industrial processing, the best concentration of methyl orange should be controlled at about 30 mg/L. .

  1. I strongly recommend to improve the English of the manuscript.

I have improved。

  1. Increase the font size and numbers in x- and y- axis in Figures 9, 10.

I have finished it.

  1. Further writing about three generations of photocatalysts is of interest.

I have finished it.

  1. What is your message with the sentence: “So our work was to prepare the third generation photocatalyst”.

I have deleted it.

  1. The authors should compare the photocatalytic performance of PFCC in this study with the other well-known photocatalysts in degradation dyes.

This is similar to the findings reported in the literature[27].

  1. For the optimizing procedure, why you chose to the optimize order presented in the manuscript (i.e., Cu2O loading, dosage, reaction time, pH, MO initial concentration.

The factors that affect the photocatalytic reaction are, the type of catalyst (titanium dioxide or other, nano or conventional scale), the type of catalytic reaction (homogeneous or heterogeneous), the basic properties of the catalyst (crystal form, particle size, dosage, specific surface, pore size, pore volume, dispersion state, etc.), light source (wavelength and intensity, etc.), photocatalytic reactor design, reaction medium (treatment object, concentration), etc. Due to the limitations of experimental conditions, this study mainly selected photocatalytic of Cu2O loading, dosage, reaction time, pH, MO initial concentration.

  1. Is there heating effect during the photocatalytic reaction under irradiation from 8W fluorescent lamp?   

Because the power of the fluorescent lamp is very small and the thermal effect is very low, the influence on the photocatalytic reaction can be ignored.

  1. The absorption spectra in UV-VIS-NIR IR range of as prepared Cu2O, PFC, PFCC samples are of interests to elucidate the roles of each material in the degradation of MO.

Cu2O is mainly photocatalysis, PFC is adsorption, PFCC is the combination of adsorption and photocatalysis in the degradation of MO.

Line 298, “reactio”   Reaction

Reviewer 4 Report

This article is interesting which could help the researcher for wastewater treatment. I would like to recommend this article for publication. However , I have the following suggestion. 

It is better if authors can correlate the effect of PH with zero point charge .  --

Author Response

Reply to  Reviewer 4

Comments and Suggestions for Authors

This article is interesting which could help the researcher for wastewater treatment. I would like to recommend this article for publication. However , I have the following suggestion. 

It is better if authors can correlate the effect of PH with zero point charge .  --

The zeta potential was not measured due to the lack of a zeta potentiometer and forgot to measure it.

Round 2

Reviewer 1 Report

I recommend acceptance of the paper

Reviewer 2 Report

Some of the concern address have been revised accordingly